# Novel Magnetic Silica-Ionic Liquid Nanocomposites for Wastewater Treatment

**DOI:** 10.3390/nano10010071

**Published:** 2019-12-28

**Authors:** Ayman M. Atta, Yaser M. Moustafa, Abdelrahman O. Ezzat, Ahmed I. Hashem

**Affiliations:** 1Chemistry Department, College of Science, King Saud University, Riyadh 11451, Saudi Arabia; ao_ezzat@ksu.edu.sa; 2Egyptian Petroleum Research Institute, Nasr City 11727, Cairo, Egypt; ymoustafa12@yahoo.com; 3Chemistry Department, College of Science, Ain Shams university, Abasia 11566, Cairo, Egypt; emyhashem2004@yahoo.com

**Keywords:** adsorbents, imidazolium cations, ionic liquid, nanocomposite, magnetite, silica nanoparticles

## Abstract

In this work, new imidazolium silica-ionic liquids doped with magnetite nanocomposites are prepared for use in the field of water purification owing to their unique properties, which can be manipulated by an external magnetic field. A silane precursor based on aminopropyltriethoxysilane (APTS) condensed with *p*-hydroxybenzaldehyde and glyoxal in an acetic acid solution is used to prepare disiloxyimidazolium ionic liquid (SIMIL). The silica composite (Si-IL) and silica-coated magnetite (Fe_3_O_4_-Si-IL) composites are prepared using the sol-gel technique. The chemical structures, morphologies, crystalline lattice structures, thermal stabilities, surface charges, surface areas, particle sizes, and magnetic characteristics of Fe_3_O_4_-Si-IL and Si-IL are investigated. The Fe_3_O_4_-Si-IL and Si-IL nanocomposites show excellent chemical adsorption capacities as 653 and 472 mg g^−1^, respectively, during times ranging 90 to 110 min when they are used as adsorbents to remove Congo red (CR) dye as a water pollutant.

## 1. Introduction

Pure and uncontaminated water is the most important basic requirement of life for all living things on Earth. More than 71% of the Earth’s surface is covered with water, but only 1% of water is drinkable as per international standards due to the presence of natural and industrial pollutants. The microorganisms, organic chemicals, and heavy metal pollutants present in water are very dangerous for human health, aquatic systems, and the environment, even in trace amounts [1,2,3]. There are different methods based on physical, chemical, and biological combinations which are used for water purification. The adsorption of pollutants onto adsorbent surfaces is known to be one of the simplest and most eco-friendly, economically feasible, and effective techniques for industrial wastewater purification [4,5,6]. There are different types of adsorbents such as natural, biomass, agricultural, and industrial waste adsorbents used as solid adsorbents for wastewater purification [4]. Moreover, advanced adsorbents based on polymers and nanomaterials such as carbon-based nanomaterials (carbon nanotubes and graphene), noble-metal-based nanomaterials, metal oxides (silica, titania, zirconia, and zinc oxides), nanocomposites, magnetic nanocomposites, dendritic polymers, and geopolymer cement have been reported for the desalination and removal of different pollutants from wastewater [5,6,7,8,9,10]. However, nanomaterials have to face some inherent technical problems when being applied to the production of large-scale adsorbents in water treatments such as aggregations, potentially adverse effects imposed on human health and ecosystems, difficult separation, and leakage into water systems. These problems can be solved by using environmentally friendly magnetic functionalized nanocomposites to emerge and integrate the merits of functional nanoparticles and vary the solid hosts of the large sizes of materials [8,9,10]. Although magnetic nanocomposites have achieved excellent results with adsorption and reuse for different cycles in the field of water treatment, some research challenges exist for their application to remove some pollutants such as toxic radioactive metals and organic and agricultural pollutants [9].

The concept of a synergistic combination of two nanoparticles into a larger one has become one of the proposed synthetic techniques used to design advanced nano-adsorbents for water treatments [11,12,13,14]. Most of these materials are based on mesoporous silica-based adsorbents [15,16,17]. The large pore sizes, high specific area, good thermal and chemical stability, low manufacture cost, and excellent selectivity towards specific pollutants elect the ordered mesoporous silica composites as excellent adsorbents for water purification and treatment [17,18,19]. Highly dispersed and magnetic silica nanoparticles with large-scale chemical or sonochemical synthesis methods have been reported [20,21,22,23,24,25]. With the large amount of literature available for the production of magnetic silica nanoparticles and with research still being published, perhaps some of the biggest challenges to overcome regard the production of greener selective magnetic silica nano-adsorbents for removal of organic pollutants; these are still a problem facing researchers. Organic cationic and anionic salts based on imidazolium, pyridinium, and quaternary ammonium cations which have been designated as ionic liquids (ILs) are widely used as greener extraction solvents for seawater desalination and industrial water and activated sludge treatments [26,27,28,29]. Poly (ionic liquid) (PIL) nanoparticles are used as flocculants for water purification [28], and hydrophobic PILs have been used for the selective separation of methylene blue (MB) dye and chromium ions from water [30]. In our previous works [31,32,33], PILs were used to prepare ordered nanoparticles with high surface charges, thermal and chemical stability, and high dispersion in aqueous systems to remove the pollutants from the water. This work aims to prepare for the first time novel porous magnetic silica nanocomposites by using aminopropyltriethoxysilane (APTS) as a reagent to prepare imidazolium ILs followed by hydrolysis and linking with magnetite in a basic medium. Moreover, the application and characterization of the produced magnetic silica-ILs as a greener, porous, and selective nanocomposite used to quickly remove MB from wastewater for different cycles are further goals of the present work.

## 2. Experimental Section

### 2.1. Materials

All chemicals used in this work were obtained from Sigma-Aldrich Chemicals Co. (Missouri, MO, USA) and used without further purification. The reagents used to prepare the silica-ionic liquids were APTS, glyoxal monohydrate (GA), acetic acid, and *p*-hydoxybenzaldehyde (PHB). The reagents used to prepare magnetite were ferric chloride hexahydrate (FeCl_3_·6H_2_O), potassium iodide (KI), and ammonium hydroxide (25%). Congo red (CR) was purchased and utilized to prepare stock solutions of 50–300 mg L^−1^. A phosphate buffer solution (H_3_PO_4_/NaH_2_PO_4_) was used to adjust the pH of aqueous solutions ranging 7–12, while acetate buffer adjusted the pH in the range 3.6–5.6.

### 2.2. Preparation Techniques

APTS (0.1 mol, 18.5 g) was mixed with 50 mL of acetic acid aqueous solution (50 vol%) in an ice bath at −4 °C. GA (0.02 mol, 1.52 g) was mixed with PHB (0.02 mol, 2.4 g) in 50 mL of acetic acid aqueous solution (50 vol%) at −4 °C to prepare the aldehyde solution. The APTS solution was added to the aldehyde solution under vigorous stirring and heated at 70 °C for 5 h. The reaction mixture was washed several times with diethyl ether to obtain a colorless organic phase. The reaction product was obtained after solvent evaporation using a rotary evaporator. The yield percentage of the reaction was 95% and the product disiloxyimidazolium ionic liquid was designated as SIMIL. Magnetite nanoparticles were synthesized via the reacting of ferric chloride with potassium iodide in the presence of alkaline solution [32]. Briefly, a solution of anhydrous FeCl_3_ (4 g, dissolved in 30 mL) was mixed and stirred with KI (1.32 g, dissolved in distilled water (5 mL)) at room temperature under an N_2_ atmosphere for one hour to keep the mixture oxygen-free. The iodine byproduct was removed by filtration of the reaction mixture. The filtrate was heated to 40 °C under vigorous stirring, and then ammonia solution (25%, 20 mL) was added dropwise to the filtrate with stirring for another 30 min at 50 °C. The Fe_3_O_4_ nanoparticles were separated from the mixture by an external magnet and washed with a water–ethanol mixture three times and then dried under vacuum. SIMIL (1 g) was dissolved into a 35% ammonia solution/ethanol mixture (100 mL, 50/50). Magnetite (1 g) was dispersed in the SIMIL solution by sonication for 5 min. Then, the reaction mixture was heated to 35 °C and stirred for 36 h until the complete hydrolysis of silicate occurred to form a magnetic silica IL (Fe_3_O_4_-Si-IL) that collected using an external magnet and was washed several times using a water:ethanol mixture (50:50). The Fe_3_O_4_-Si-IL yield percentage of the reaction was 98.9%. The Si-IL was prepared under the same conditions in the absence of magnetite with the reaction yield percentage being 97.6%.

### 2.3. Characterization of the Prepared Composites

The chemical structures of the prepared silicate and SIMIL, Fe_3_O_4_-Si-IL, and Si-IL composites were elucidated using Fourier transform infrared analysis (Nicolet Magna 750 FTIR spectrometer using KBr, Newport, NJ, USA) and hydrogen nuclear magnetic resonance (^1^HNMR) spectroscopy recorded on a 400 MHz 400 MHz Bruker Avance DRX-400 spectrometer (Toronto, ON, Canada). Their thermal stability was determined using thermogravimetric analysis (TGA; NETZSCH STA 449 C instrument, New Castle, DE, USA) under an N_2_ atmosphere with heating rate 10 °C min^−1^. X-ray powder diffraction (X’Pert, Philips, Amsterdam, Netherlands, using CuKa radiation of wavelength λ = 1.5406 °A) was utilized to identify the crystal lattice structures and diffraction patterns of the Fe_3_O_4_-Si-IL and Si-IL. The particle size, polydispersity index (PDI), and zeta potential of Fe_3_O_4_-Si-IL and Si-IL were determined using dynamic light scattering (DLS) (Malvern Instrument Ltd., London, UK) at different pHs. The surface morphologies of Fe_3_O_4_, Fe_3_O_4_-Si-IL, and Si-IL were studied by transmission electron microscopy (TEM JEOL JEM-2100 F at acceleration voltage of 200 kV, JEOL, Tokyo, Japan) and scanning electron microscopy (JEOL JXA-840A) at 200 kV. The magnetic properties of the Fe_3_O_4_ and Fe_3_O_4_-Si-IL were measured using a vibrating sample magnetometer (VSM; USALDJ9600-1; LDJ Electronics, Troy, MI, USA) at room temperature. Brunauer-Emmett-Teller (BET) was used to determine the surface area, the pore volume, and the pore-size distribution of Fe_3_O_4_-Si-IL and Si-IL powders by the nitrogen adsorption–desorption process using the adsorption data in the relative pressure (*P*/*P*_0_) range of 0.05–0.3 using a Belsorp-mini II (BEL; Tokyo, Japan).

### 2.4. CR Adsorption

A UV−visible spectrophotometer (Shimadzu UV-1208 model; Canby, OR, USA) was used to determine the adsorption characteristics of Fe_3_O_4_-Si-IL and Si-IL adsorbents to remove CR from water. A UV–visible spectrophotometer was used to determine the CR absorbance at wavelengths of 496, 495, and 564 nm at pHs of 9, 7, and 4, respectively. The adsorption capacities of CR dye at equilibrium *qe* (mg g^−1^) and the adsorption efficiency *E* (%) were determined using *qe* = (Co − Ce) × V/m and *E* (%) = (Co − Ce) × 100/Co, where Co, Ce, V and m are the dye concentration in aqueous solutions at time zero at equilibrium (mg L^−1^), the volume of the aqueous solution (L), and the adsorbent mass (g), respectively. The optimum conditions such as effective Fe_3_O_4_-Si-IL and Si-IL adsorbent weights, the pH of aqueous solutions, and contact times on the CR maximum adsorption capacity were determined at constant room temperature, and ionic strength 0.01 M and a CR of 100 mg g^−1^. The optimum conditions of pH, contact time, and weight of Fe_3_O_4_-Si-IL and Si-IL adsorbents are used to study their adsorption isotherms and kinetics to remove CR from the aqueous solution and to purpose the predominant adsorption removal mechanism. Temperatures of aqueous solutions ranging 298 to 313 K were used to investigate thermodynamic parameters such as standard Gibbs energy (∆*G*_o_; in kJ mol^−1^), enthalpy (∆*H*_o_; in kJ mol^−1^), and entropy (∆*S*_o_; in J mol^−1^ K) of the CR adsorption on the Fe_3_O_4_-Si-IL and Si-IL adsorbent surfaces.

The CR was desorbed from CR-loaded SI-IL and Fe_3_O_4_-Si-IL by washing with water and stirring in 25 mL of water followed by stirring in 25 mL of 0.1 M of NaOH solution for 3 h. The concentration of CR was determined as reported in the adsorption section and the desorption efficiency (DE%) of CR was calculated as the ratio of the desorption capacity (*q_de_*) and adsorption capacity (*q_e_*). The reusability of the regenerated SI-IL and Fe_3_O_4_-Si-IL was checked using seven cycles of consecutive adsorption–desorption studies.

## 3. Results and Discussion

The present work aimed to prepare novel hybrid silica nanoparticles linked with imidazolium ILs to control their sizes, shapes, surface charges, and thermal and chemical stability. In a previous report imidazolium ILs were prepared by condensation of alkyl amines with GA and PHB under acetic acid conditions [32]. In this respect, APTS was selected as a silane precursor to condense with PHB and GA under acetic acid conditions to produce SIMIL as represented in Scheme 1. The triethoxysilane group at the terminal of SIMIL can hydrolyze with tetraethoxysilane (TEOS) or TEOS and magnetite in the presence of ammonia to produce an Si-IL and Fe_3_O_4_-Si-IL hybrid. Ammonia was selected to hydrolyze SIMIL due to it increasing its solubility as an aprotic IL under alkaline conditions more than acidic conditions [34]. The presence of hydroxyl groups on the surface of magnetite [32] facilitates its linking with silica colloids during the hydrolysis of SIMIL under basic conditions.

### 3.1. Characterization

The chemical structure of SIMIL was investigated using the ^1^HNMR spectrum represented in Figure 1. The formation of imidazolium cations is elucidated by the appearance of a singlet peak at 6.8 ppm related two hydrogen olefin protons. The disappearance of the aldehyde protons of *p*-hydroxybenzaldehyde (HBA) at 9.3 ppm and appearance of peaks at 7.6, 7.3, and 2.3 ppm attributed to four protons of *p*-substituted phenyl rings and phenolic –OH, respectively, confirm the presence of phenol in the chemical structure of SIMIL. The appearance of peaks at 4.19 (m, 2H, CH_2_-N^+^), 3.64 (m, 2H, CH_2_-N), 3.00 (m, 12H, (CH_2_-O)_3_) 1.89 (m, 8H, 2 (CH_2_)_2_), 1.62 (s, 3H, CH_3_COO^−^), and 0.55 ppm (m, 18H, 2 (CH_3_)_3_) elucidate the incorporation of APTS within the chemical structure of SIMIL (Figure 1).

The FT-IR spectra of Si-IL and the Fe_3_O_4_-Si-IL hybrid represented in Figure 2a,b were used to confirm the formation of a silica colloid and the Si-Fe_3_O_4_ hybrid and to detect their surface structure. There are three bands able to be observed at 580, 1560, and 3607 cm^−1^ in the spectrum of Fe_3_O_4_-Si-IL (Figure 2b) which correspond to the Fe-O vibration, C=NH stretching of imidazole, and O-H stretching vibration, respectively. The formation of Si-O-Si can be confirmed from the formation of broad bands at 1055–1070 cm^−1^ and 793–938 cm^−1^ which can be attributed to its asymmetric and symmetric stretching vibration, as observed in the spectra of Si-IL and Fe_3_O_4_-Si-IL (Figure 2a,b). It was noticed that the intensity of the O-H stretching vibration increased after hydrolysis of SIMIL to form colloid Si-IL, which indicated an increase in the content of Si-OH concentration. Moreover, the intensity of the O-H band at 3400 cm^−1^ can be seen to have been reduced in the spectrum of Fe_3_O_4_-Si-IL, confirming the linking of the hydroxyl groups surrounded on the magnetite surfaces with the hydroxyl groups of Si-OH during the hydrolysis of SIMIL in the presence of magnetite nanoparticles (NPs). These results confirm that the magnetite NPs were coated with silica via the hydrolyzing of TEOS with SIMIL.

The thermal stability and silica or magnetite contents of SIMIL, Si-IL, Fe_3_O_4_ NPs, and Fe_3_O_4_-Si-IL hybrid were evaluated using TGA, as represented in Figure 3. The data show that SIMIL was degraded at 325 °C and that the incorporation of magnetite and the formation of silica increased the lost weight (%) by up to 8 wt.% at a temperature starting from 90 °C. This wt.% can be attributed to the humidity and the presence of water contaminated with magnetite, silica colloid and imidazolium ILs [35]. Accordingly, the starting degradation temperatures for Si-IL and Fe_3_O_4_-Si-IL, determined as the temperature at which the NPs lost approximately 10 wt.%, were 300 and 425 °C, respectively. The strong interaction and linking of magnetite with silica in the Fe_3_O_4_-Si-IL hybrid improved its thermal stability due to its reducing the plasticization effect of IL [27]. The residual contents of SIMIL, Si-IL, and the Fe_3_O_4_-Si-IL hybrid after decomposition above 750 °C, determined from their thermograms (Figure 2), can be seen to be 16, 57, and 76 wt.%, respectively. These data determined the pure silica and magnetite contents to be 41 and 18 wt.%, respectively, after subtracting the water and IL contents.

The particle size distributions and surface charges of Si-IL and the Fe_3_O_4_-Si-IL hybrid were evaluated using DLS measurements and have been summarized in Figure 4a–c. The relationship between the surface charges of Si-IL and the Fe_3_O_4_-Si-IL hybrid and the pH of their aqueous dispersion has been represented in Figure 5 and was used determine their pHs at the isoelectric point (zero surface charges). PDI data confirm the formation of uniform particle sizes when the values decreased below 0.7 and that the particles were more uniform when the values were close to 0.1 [36]. The data of particle sizes of Si-IL (Figure 4a–c) indicate that the hydrolysis of TEOS with SIMIL in basic medium produced silica nanoparticles with sizes ranging from 10.5 to 22.5 nm without the formation of aggregates in aqueous solutions which had different pHs. The presence of Fe_3_O_4_ during the hydrolysis of TEOS with SIMIL in basic medium produced a Fe_3_O_4_-Si-IL hybrid with the formation of small aggregates at 5.8 nm via increasing the particle sizes of the silica nanoparticles in the absence of Fe_3_O_4_ from 10.9 (Si-IL) to 29.6 nm in the presence of Fe_3_O_4_ (Fe_3_O_4_-Si-IL). The presence of small particle sizes at 5.8 nm may refer to the presence of free magnetite nanoparticles that non-linked with silica during the hydrolysis. The improvement of dispersity of Fe_3_O_4_-Si-IL with reduction of PDI values and increasing pH (Figure 4a–c) may be attributed to an increase in their surface charges from +10.9 to −29.7 mV (Figure 5) and repulsive forces between nanoparticles. The negative surface charges of both Si-IL and the Fe_3_O_4_-Si-IL hybrid indicate the presence of hydroxyl groups at the surface of the silica nanoparticles [37]. Moreover, the positive surface charges of both Si-IL and Fe_3_O_4_-Si-IL in acidic medium can be attributed to the positive charges of the imidazolium cations. Consequently, it can be concluded that the imidazolium cations oriented as a shell at the surface of Si-IL and the Fe_3_O_4_-Si-IL hybrid in acidic medium and that their hydroxyl groups oriented as a shell in the basic medium. It can also be noted that the isoelectric points of Si-IL and the Fe_3_O_4_-Si-IL hybrid were obtained at pH values of 4.8 and 6.6, respectively (Figure 5). This means that the linking of the hydroxyl group of magnetite during the hydrolysis of silica facilitates the orientation of imidazolium cations into the core and their hydroxyl groups as a shell, even in a slightly acidic and neutral aqueous solution. The zeta potential of Fe_3_O_4_-Si-IL (Figure 5) was less negative when the pH of the aqueous solution increased from 7 to 12 to elucidate the repulsive forces occurring between the magnetite nanoparticles and negative charges of the basic medium besides reducing the interaction of the imidazolium cations of Fe_3_O_4_-Si-IL with the basic medium. This speculation can be elucidated also when observing the increase in particle sizes of Si-IL from 12.5 to 22.5 nm, as the pH changed from 7 to 9 (Figure 4b,c), when compared to the slight increasing of Fe_3_O_4_-Si-IL particle sizes from 24.4 to 33.82 nm. The greater increasing of Si-IL compared to Fe_3_O_4_-Si-IL particle sizes was able to confirm the higher interaction of imidazolium cations of Si-IL particles with alkaline solution when compared with Fe_3_O_4_-Si-IL.

The morphologies of Fe_3_O_4_, Si-IL, and the Fe_3_O_4_-Si-IL hybrid were able to be evaluated by high resolution transmittance electron microscope (HR-TEM) micrographs, which are represented in Figure 6a–c, respectively. The data show that spherical nanoparticles were formed in the cases of Si-IL and the Fe_3_O_4_-Si-IL hybrid (Figure 6b,c). It can also be observed that a porous silica shell was formed in both Si-IL and the Fe_3_O_4_-Si-IL hybrid (top of Figure 6b,c) on imidazolium IL and magnetite, respectively, as a core. The hydrolysis of SIMIL with TEOS in the presence of ammonia converted the phenol groups of SIMIL to its phenoxy ammoniate ion, increasing the solubility of TEOS and increasing the rate of hydrolysis of SIMIL with TEOS. Consequently, the uniform silica shell increased the repulsive forces among particles to produce highly dispersed uniform Si-IL (Figure 6b). The presence of rhombic magnetite nanoparticles (Figure 6a) changed the polarity of the Fe_3_O_4_-Si-IL particles to control the interactions of the particles and the thickness of the core-shell morphologies [38].

The lattice structures of Si-IL and the Fe_3_O_4_-Si-IL hybrid were investigated using their XRD diffractograms, as represented in Figure 7a,b, respectively. The Fe_3_O_4_-Si-IL hybrid diffractogram indicates that the corresponding peaks and lattice structure magnetite (Figure 7b) used during the hydrolysis of SIMIL with TEOS was pure without oxidation to other iron oxide impurities such as maghemite or hematite [39]. The presence of a broad peak at 2-theta 15–22° confirms the formation of amorphous porous silica shells in both Si-IL and Fe_3_O_4_-Si-IL, while the multiple high angle peaks relate to the magnetite core of Fe_3_O_4_-Si-IL (Figure 7a,b).

The magnetic properties of the Fe_3_O_4_-Si-IL hybrid were determined using a VSM hysteresis loop (Figure 8). The magnetic properties of Fe_3_O_4_-Si-IL, including saturation magnetization (Ms; this represents the magnetization of the material at a higher applied value of an external magnetic field H), coercivity (Hc; this measures the reverse field required to reach the magnetization to zero after saturation), and remnant magnetization (Mr; this evaluates the remaining magnetization after removal of the external magnetic field), were found to be 35.3 emu/g, 20.71 G, and 0.38 emu/g, respectively. The Ms, Hc, and Mr of the prepared magnetite NPs had been determined in a previous work [40] as 78.6 emu/g, 8.55 G, and 0.15 emu/g, respectively. The lower Ms value of Si-IL indicates that the magnetite was encapsulated or coated into Si-IL, which reduced its magnetization. The low Hc value of Fe_3_O_4_-Si-IL confirms that the prepared material possesses superparamagnetic properties.

The pore structure and surface area of Si-IL and Fe_3_O_4_-Si-IL were estimated from nitrogen adsorption–desorption isotherms at 77 K, which are represented in Figure 9. The samples were pretreated at a temperature of 353 K to remove any water or water humidity from the sample pores [41]. The samples were then heated under vacuum up to 423 K before measuring their surface area and pore sizes. The sorption isotherms of Si-IL and Fe_3_O_4_-Si-IL (Figure 9) gave rise to type I. The BET surface area (*S_BET_*; m^2^ g^−1^), pore sizes (*D*; nm), and pore volume (*V_total_*; cm^3^ g^−1^) of Si-IL were found to be 64, 10.56, and 0.0869, respectively. The *D*, *S_BET_*, and *V_total_* values of Fe_3_O_4_-Si-IL were found to be 92 m^2^ g^−1^, 14.63 nm, and 0.1729 cm^3^ g^−1^, respectively. These data agree with the DLS data (Figure 4a–c) and TEM micrographs (Figure 6b,c) and indicate that the pore sizes increased in the case of Fe_3_O_4_-Si-IL due to the linking of magnetite with silica to activate the silica surfaces and increase their surface area. These data confirm the formation of the porous structures of Si-IL and Fe_3_O_4_-Si-IL, which enhance their application as an adsorbent.

### 3.2. Application of Adsorbents for CR Removal

The optimum pHs with which to remove CR from their 20 mL aqueous solution at a concentration of 100 mg L^−1^ were determined from the relationship of removal efficiency (*E*%) versus pH, which has been plotted in Figure 10, at constant room temperature, ionic strength 0.01 M, and 4 mg of the adsorbents. The optimum pH 4 achieved a higher *E*% of both Si-IL and Fe_3_O_4_-Si-IL of 94.2 and 99.9%, respectively. It may also be noted that *E*% decreased dramatically with increasing pH of the aqueous solution to neutral and basic solutions. This observation can be correlated to the reaction of hydroxyl phenolic of SIMIL in the basic medium to produce anionic phenoxide, which was repulsed by the negative charges of CR moieties that increased in the basic or neutral aqueous solution to enhance the electrostatic attraction between the imidazolium site of the adsorbents and the anionic CR dye in the acidic medium [42]. The encapsulation of magnetite inside the shell of Fe_3_O_4_-Si-IL protects it from leaching or decomposition in acidic solution [43].

The optimum weights of the Si-IL and Fe_3_O_4_-Si-IL adsorbents to remove 100 mg L^−1^ of CR from 20 mL of aqueous solution at pH 4 and room temperature were determined from the relationship between *E*% or *q* (mg g^−1^) and adsorbent weights (mg), as represented in Figure 11a,b, respectively. The data show that the optimum weights of the Si-IL and Fe_3_O_4_-Si-IL adsorbents which achieved a higher *E*% of CR removal were 3.7 and 3 mg, respectively. Moreover, the *q* and *E*% of the Fe_3_O_4_-Si-IL adsorbent (Figure 11b) at its optimum weight were 650 mg g^−1^ and 98.8%, respectively. The Si-IL (Figure 11a) adsorbent displayed a lower *q* and *E*% at its optimum weight, and *q* and *E*% were found to be 476 and 90%, respectively. These data indicate the higher adsorption capacity and removal efficiency at lower optimum weight of Fe_3_O_4_-Si-IL compared to Si-IL due to its higher surface area, pore volume, and diameter (BET measurements; Figure 9).

The effect of contact times on the CR maximum adsorption capacity (*q*; mg g^−1^) of both Fe_3_O_4_-Si-IL and Si-IL at pH 4 and room temperature and their optimum weights was investigated. The data confirm that Fe_3_O_4_-Si-IL and Si-IL achieved maximum q values of 653 and 472 mg g^−1^ at contact times of 110 and 90 min, respectively. Moreover, it may be noted that Fe_3_O_4_-Si-IL showed greater *q* values during all contact times to reach the equilibrium. By comparing our work with previous works aimed at removing CR from aqueous solution [44,45,46,47], it was able to be observed that the present system achieved q values ranging 472 to 653 during times ranging 90 to 110 min, which are higher than other systems which achieved *q* values ranging from 950 to 500 mg g^−1^ during a period of 10 h.

### 3.3. Adsorption Isotherms

The homogeneity and heterogeneity of the Fe_3_O_4_-Si-IL and Si-IL adsorbent surfaces beside the formation of the CR monolayer and multilayers on the adsorbent surfaces were confirmed using the best adsorption isotherms models of Langmuir and Freundlich. The Langmuir and Freundlich adsorption models can be represented using the equations (*C*_e_/*q*_e_) = [(1/*Q*_max_*K*_l_) + (*C*_e_/*Q*_max_)] and log(*q*_e_) = log(*K*_f_) + [(1/*n*) log(*C*_e_)]. The constants *n* (in g L^−1^), *K*_l_ (in L mg^−1^), and *K*_f_ (in (mg g^−1^)(L mg^−1^)^(1/*n*)^) are the empirical constant, Langmuir constant, and Freundlich constant, respectively. The adsorption parameters of the Langmuir and Freundlich equations related to the Fe_3_O_4_-Si-IL and Si-IL adsorbents are listed in Table 1. The linear relationships of both the Fe_3_O_4_-Si-IL and Si-IL adsorbents with the highest linear coefficient (*R*^2^) (Table 1) reveal that the present system obeyed the Langmuir more than the Freundlich adsorption model. This means that both the Fe_3_O_4_-Si-IL and Si-IL adsorbents had homogenous surfaces and formed a monolayer of CR adsorbate on their surfaces. Moreover, the good agreement between the theoretical adsorption capacity *Q_max_* (mg g^−1^) and experimental value *q_e_* using the Langmuir model proves that the adsorption process obeyed the Langmuir model.

The essential characteristic of the Langmuir isotherm can be expressed in terms of a dimensionless constant separation factor (RL; also called the equilibrium parameter) which is given by the equation *RL* = 1/(1 + KlCo), where Co (mg/L) is the highest initial CR dye concentration (mg L^−1^) [48]. The *RL* values of the Fe_3_O_4_-Si-IL and Si-IL adsorbents were found to be 0.009855 and 0.009853, respectively, which indicates that the adsorption of CR on their surfaces was favorable because the 0 < *RL* < 1. Additionally, the Fe_3_O_4_-Si-IL and Si-IL adsorbents showed lower *RL* values at higher initial CR concentrations, confirming that the adsorption was more favorable at a higher concentration. This degree of favorability is generally related to the irreversibility of the system, giving a qualitative assessment of the interactions of Fe_3_O_4_-Si-IL and Si-IL towards CR. The *RL* tended toward zero, confirming the completely ideal irreversible adsorption case of CR on the Fe_3_O_4_-Si-IL or Si-IL surfaces rather than unity (which represents a completely reversible case).

### 3.4. Adsorption Kinetics

The adsorption rates of Fe_3_O_4_-Si-IL and Si-IL adsorbents to remove CR from aqueous solutions were able to be estimated and analyzed using pseudo first-order and pseudo second-order models [49] and have been tabulated in Table 2. The higher correlation coefficient (*R*^2^) and the agreement of the calculated *q*_e,cal_ values of the fitted curves obtained by the pseudo second-order kinetic model with the experimental *q*_e,exp_ values prove that the pseudo second-order model is the predominant means by which to describe the adsorption process. These data mean that the adsorption capacities of the Fe_3_O_4_-Si-IL and Si-IL adsorbents are more dependent on the CR concentrations. By comparing the adsorption rate constant (*K*_2_; Table 2) of the present system with other IL systems reported in the literature [44,45,46,47], it was able to be found that the present systems for both the Fe_3_O_4_-Si-IL and Si-IL adsorbents have a higher and faster rate of removal of CR as compared to other PILs. Moreover, the removal rate of Fe_3_O_4_-Si-IL was found to be two times faster than Si-IL (*K*_2_; Table 2). This means the adsorption capacity was more dependent on the surface adsorbate amount [4].

The effect of temperature on thermodynamic parameters such as standard Gibbs energy (∆*G*_o_; in kJ mol^−1^), enthalpy (∆*H*_o_; in kJ mol^−1^), and entropy (∆*S*_o_; in J mol^−1^ K) of CR adsorption on the Fe_3_O_4_-Si-IL and Si-IL adsorbent surfaces were investigated and calculated using the equations ∆*G*_o_ = −*RT* ln(*C*_e_*A*/*C*_e_) and log(*C*_e_*A*/*C*_e_) = ∆*S*_o_/2.303*R* − ∆*H*_o_/2.303*RT*, where *C*_e_*A*, *R*, and *T* are the adsorbent concentration (in mg L^−1^), gas constant (8.314 J mol^−1^ K^−1^), and the aqueous solution temperature (in K), respectively. The values of ∆*G*_o_, ∆*H*_o_, and ∆*S*_o_ of Fe_3_O_4_-Si-IL and Si-IL have been given in Table 3. The equilibrium concentration constant (*K*_c_) was able to be calculated from the relation (*C*_e_*A*/*C*_e_) for Fe_3_O_4_-Si-IL and Si-IL and has been plotted in Figure 12. It may be noted from the data in Table 3 that the higher negative value of ∆*G*_o_ for Fe_3_O_4_-Si-IL compared to Si-IL with increasing temperature highlights the spontaneous nature of the CR dye adsorbed on the Fe_3_O_4_-Si-IL adsorbent at a higher rate than Si-IL. The positive values of ∆*H*_o_ and ∆*S* confirm that the adsorption of the CR adsorbate on the Fe_3_O_4_-Si-IL and Si-IL surfaces was endothermic, and also indicates an increase in the degrees of freedom of the CR species on the surface of the adsorbent with increasing temperature [50]. Moreover, these values show that an increase in the concentration of the CR dye molecules at the solid–liquid interface increased its adsorption onto the Fe_3_O_4_-Si-IL and Si-IL surfaces.

The previous data were used to confirm the proposed adsorption mechanism represented in Scheme 2 for removal of CR from the aqueous solution by using Fe_3_O_4_-Si-IL and Si-IL adsorbents. In this respect, the porous, uniform, and ordered surfaces of Fe_3_O_4_-Si-IL and Si-IL can be seen to have facilitated the formation of a monolayer of CR on their surfaces, as elucidated from the Langmuir isotherm (Table 1). The CR diffused into the porous structure of Fe_3_O_4_-Si-IL and Si-IL, which was confirmed via TEM (Figure 6), BET measurements (Figure 9), and the fact that the adsorption capacity increased by ion exchange with the chemical binding. Moreover, SIMIL is shown to play a significant role in increasing the CR adsorption capacity. It is responsible for the formation of a strong hydrogen bond, ion exchange CR anions, and CH_3_COO^–^ counter-ions, and the electrostatic interaction between the imidazolium cations of the adsorbent and the sulfonate anion of the adsorbate are responsible for the chemisorption nature of Fe_3_O_4_-Si-IL and Si-IL toward CR dye [51,52].

A comparison between the adsorption efficiencies of the Fe_3_O_4_-Si-IL and Si-IL nanocomposites and other sorbents was carried out, as displayed in Table 4. The prepared nanocomposites removed CR dye from water with higher capacities and in shorter times compared to other adsorbents [53,54,55]. The porous structures of Si-IL and Fe_3_O_4_-Si-IL enabled them to adsorb CR dye with a high capacity. In addition, the imidazolium cation in the silicate network facilitated the CR adsorption through an electrostatic interaction mechanism.

Seven cycles of the desorption–sorption experiments to reuse of Fe_3_O_4_-Si-IL and Si-IL were performed in basic solution, as reported in the experimental section. The *DE*% of Fe_3_O_4_-Si-IL and Si-IL were determined. The data indicate that the desorption of CR was completed in basic medium and could not be desorbed in neutral medium, which confirms that the linking between CR and Fe_3_O_4_-Si-IL and Si-IL is by an ion exchange strong force [56]. The magnetic properties of Fe_3_O_4_-Si-IL during seven cycles confirmed the bonding of magnetite with silica and the good protection of SIMIL, which prevents further oxidation or leaching of magnetite. The *DE*% of Si-IL shows that the CR dye desorption efficiencies decreased by 75% after the fifth cycle. This could be referred to as the demolition of the electrostatic attraction between the Si-IL and CR molecules.

## 4. Conclusions

In this work, a new aprotic disiloxyimidazolium ionic liquid SIMIL was hydrolyzed with TEOS or TEOS and magnetite in the presence of ammonia to produce Si-IL and a Fe_3_O_4_-Si-IL hybrid. The pure silica and magnetite contents of the Fe_3_O_4_-Si-IL hybrid were 41 and 18 wt.%, respectively. The presence of Fe_3_O_4_ during the hydrolysis of TEOS with SIMIL increased the particle sizes of silica nanoparticles from 10.9 (Si-IL) to 29.6 nm (Fe_3_O_4_-Si-IL). The imidazolium cations oriented as a shell at the surface of Si-IL and the Fe_3_O_4_-Si-IL hybrid in acidic medium and their hydroxyl groups were oriented as a shell in the basic medium. The isoelectric points of Si-IL and the Fe_3_O_4_-Si-IL hybrid were obtained at pH 4.8 and 6.6, respectively. This confirms that the linking of the hydroxyl group of magnetite during the hydrolysis of silica facilitated the orientation of imidazolium cations into the core and their hydroxyl groups as a shell, even in slightly acidic and neutral aqueous solution. The uniform silica shell increased the repulsive forces among particles to produce a highly dispersed uniform Si-IL. The presence of rhombic magnetite nanoparticles changed the polarity of the Fe_3_O_4_-Si-IL particles to control the interactions of the particles and the thickness of the core–shell morphologies. BET data confirmed the formation of the porous structures of Si-IL and Fe_3_O_4_-Si-IL. The Fe_3_O_4_-Si-IL and Si-IL nanocomposites showed excellent chemical adsorption capacities of 653 and 472 mg g^−1^, respectively, during times ranging from 90 to 110 min when they were used as adsorbents to remove CR dye as a water pollutant.

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
