# Peer review of "Novel Magnetic Silica-Ionic Liquid Nanocomposites for Wastewater Treatment"

_nanomaterials, 2019, doi:10.3390/nano10010071_

Round 1

Reviewer 1 Report

Observation:

The title of manuscript is clear and reflect the content.

The manuscript adheres to the journal's standards, but must be improve.

The authors must underline the major findings of their work and explain how the use of their proposed procedures and resulted materials represents a progress to other similar published papers. Please proved comparatively table.

The Abstract must be rewrite and completed with data about adsorption, efficiency, kinetic etc. Abstract section should refer to the study findings, methodologies, discussion as well as conclusion

The key words permit found article in the current registers or indexes.

In the Introduction is clearly described the state of the art of the investigated problem.

The paper was written in standard, grammatically correct English, small corrections are necessary.

Please verify:

Scale for adsorption capacities in Figures.

All chemicals used in this work were obtained from Sigma-Aldrich chemicals….

Please complete with u.m. For example: q values ranged from 472 to 653 during times ranged …..

The desorption studies is very important, if is possible present separately not in   …..3.3. Adsorption isotherms and kinetics

The equations in text are hard to understand.

The Conclusion must be rewrite. The authors must present major results of the research.

Please provide minimum 2 references from this journal (last years), for demonstrated that manuscript is in Nanomaterials Journal topic.

The references are from the last years.

Please verify all references; the authors guide is not respected.

Author Response

The manuscript adheres to the journal's standards, but must be improve.

The manuscript was revised and enhanced.

The authors must underline the major findings of their work and explain how the use of their proposed procedures and resulted materials represents a progress to other similar published papers. Please proved comparatively table.

A comparative table was attached to indicate the superior of the prepared materials when comparing with other materials.

The Abstract must be rewrite and completed with data about adsorption, efficiency, kinetic etc. Abstract section should refer to the study findings, methodologies, discussion as well as conclusion

The abstract was edited as recommended.

The paper was written in standard, grammatically correct English, small corrections are necessary.

The grammatical corrections were carried out for all the manuscript.

Please complete with u.m. For example: q values ranged from 472 to 653 during times ranged …..

It was completed in line 18 and 19.

The desorption studies is very important, if is possible present separately not in   …..3.3. Adsorption isotherms and kinetics

They were written separately.

The Conclusion must be rewrite. The authors must present major results of the research.

The conclusion was edited as recommended.

Please provide minimum 2 references from this journal (last years), for demonstrated that manuscript is in Nanomaterials Journal topic.the references are from the last years.Please verify all references; the authors guide is not respected.

Reference 4,5 which are related to nanomaterials journal were added.

Reviewer 2 Report

In this manuscript, disiloxyimidazolium ionic liquid (SIMIL) and silica-coated magnetite (Fe3O4-Si-IL) were prepared and evaluated for Congo Red (CR) adsorption. The formation of SIMIL and Fe3O4-Si-IL were verified and they were characterized with TGA, XRD, TEM, PSD, N2 adsorption/desorption, and surface charge. The CR adsorption was investigated at different pH, different dose. The adsorption isotherms, kinetics, thermodynamics were also investigated.

Overall, the topic is interesting, the manuscript was well organized. However, parts of the manuscript are hard to understand and several terms were misunderstood to lead strange discussion/conclusion. In addition, grammatical errors and unclear and/or awkward statements are frequently found.

Therefore, I recommend this manuscript for major revision, since this manuscript should be significantly improved before publication.

Additional specific comments for the improvement follow;

Line(s) 18. Please specify the adsorption, such as adsorption capacity and conditions.

Line(s) 18. Please delete “data”.

Line(s) 28-29. Please revise the sentence “There are many techniques were used for water purification”.

Line(s) 119-131. This can be separated in a section of CR adsorption experiments.

Line(s) 119-131. Please provide the experimental conditions of isotherm and kinetics.

Line(s) 156-168. Please provide the FTIR spectra either in the manuscript or in a separate supplementary material.

Line(s) 194-196. I think the PDI means the dispersity of particle sizes, which is in indicator of size uniformity. So, I do not think that the PDI is an indicator of the dispersion/stabilization of particles in a liquid.

Line(s) 209, Figure 4. I think the authors should explain why the zeta potential of Fe3O4-Si-IL increased when the pH was increased from 7 to 12.

Line(s) 223-225, Figure 5. Please provide elemental maps to ensure formation of the core-shell structures.

Line(s) 235-244. Please provide the VSM results in the manuscript or in a separate supplementary material.

Line(s) 294-309. I think it is worth trying Tempkin and Redlich–Peterson or Sips isotherms to have more information about the adsorption phenomena.

Line(s) 306-309. With the results of the Langmuir isotherm, more discussion is possible. For example, a separation factor (Webber & Chakkravorti, 1974, AlChE J. 20, 228–238) can be obtained to determine whether the adsorption is favorable or not.

Line(s) 314-317. Please discuss the meaning of the better fit to pseudo-second order adsorption kinetic model. Generally, it means the adsorption capacity is more dependent on the surface adsorbate amount.

Author Response

Reviewer 2

However, parts of the manuscript are hard to understand and several terms were misunderstood to lead strange discussion/conclusion. In addition, grammatical errors and unclear and/or awkward statements are frequently found.

Therefore, I recommend this manuscript for major revision, since this manuscript should be significantly improved before publication.

Additional specific comments for the improvement follow; 

Line(s) 18. Please specify the adsorption, such as adsorption capacity and conditions.

Answer: values were added

Line(s) 18. Please delete “data”.

Answer: data deleted

Line(s) 28-29. Please revise the sentence “There are many techniques were used for water purification”.

Answer: It was mo0dified as There are different methods based on physical, chemical and biological combinations were used for water purification. The adsorption of pollutants on the adsorbent surfaces is selected as one of the simplest, eco-friendly, economically feasible and effective technique for the industrial wastewater purification

Line(s) 119-131. This can be separated in a section of CR adsorption experiments.

Answer: new section added and the measurements conditions inserted in this section.

Line(s) 119-131. Please provide the experimental conditions of isotherm and kinetics.

Answer: new section added and the measurements conditions inserted in this section.

Line(s) 156-168. Please provide the FTIR spectra either in the manuscript or in a separate supplementary material.

Answer. FTIR spectra inserted as Figure 2a and b.

Line(s) 194-196. I think the PDI means the dispersity of particle sizes, which is in indicator of size uniformity. So, I do not think that the PDI is an indicator of the dispersion/stabilization of particles in a liquid.

Answer: I agree the reviewer comment and new sentence clarified the meaning as: The polydispersity index (PDI) data confirm the formation of uniform particle sizes when their values decreased below 0.7 and they are more uniform when their values closed to 0.1.

Line(s) 209, Figure 4. I think the authors should explain why the zeta potential of Fe3O4-Si-IL increased when the pH was increased from 7 to 12.

Answer: The zeta potential of Fe3O4-Si-IL (Figure 5) was less negative when the pH of aqueous solution was increased from 7 to 12 to elucidate the repulsive forces occurred between the magnetite nanoparticles and negative charges of the basic medium beside reduces the interaction of imidazolium cations of Fe3O4-Si-IL with basic medium. This speculation elucidated also from increasing the particle sizes of Si-IL from 12.5 to 22.5 nm as the pH changed from 7 to 9 (Figure 4b and c) when compared to slight increasing of Fe3O4-Si-IL particle sizes from 24.4 to 33.82 nm. The higher increasing of Si-IL more than Fe3O4-Si-IL particle sizes confirm the higher interaction of imidazolium cations of for Si-IL particles with alkaline solution more than Fe3O4-Si-IL.

Line(s) 223-225, Figure 5. Please provide elemental maps to ensure formation of the core-shell structures.

Answer: It is not available but I added higher resolution particles on the top of each TEM figured.

Line(s) 235-244. Please provide the VSM results in the manuscript or in a separate supplementary material.

Answer. VSM inserted as figure 8.

Line(s) 294-309. I think it is worth trying Tempkin and Redlich–Peterson or Sips isotherms to have more information about the adsorption phenomena.

Answer: The samples were pretreated at a temperature of 353K to remove any water or water humidity from samples pores. The samples were heated under vacuum up to 423K before measuring their surface area and pore sizes. The sorption isotherms of Si-IL, and Fe3O4-Si-IL gave rise to type I. The isother used yo determine BET surface area (SBET; m2.g-1), pore sizes (D; nm), and pore volume (Vtotal; cm3.g-1).

Line(s) 306-309. With the results of the Langmuir isotherm, more discussion is possible. For example, a separation factor (Webber & Chakkravorti, 1974, AlChE J. 20, 228–238) can be obtained to determine whether the adsorption is favorable or not.

The essential characteristic of the Langmuir isotherm can be expressed in terms of a dimensionless constant separation factor (RL, also called equilibrium parameter) which is given by the equation: RL = 1 / (1+KlCo); where Co (mg/L) is the highest initial CR dye concentration (mg.L-1). The RL values of the Fe3O4-Si-IL and Si-IL adsorbents are 0.009855 and 0.009853, respectively which indicate that the adsorption of CR on their surfaces is favorable because the 0<RL<1. Also the Fe3O4-Si-IL and Si-IL adsorbents show lower RL values at higher initial CR concentrations to confirm that the adsorption was more favorable at higher concentration. The degree of favorability is generally related to the irreversibility of the system, giving a qualitative assessment of the interactions of Fe3O4-Si-IL or Si-IL towards CR. The RL tended toward zero to confirm the completely ideal irreversible adsorption case of CR on the Fe3O4-Si-IL or Si-ILsurfaces rather than unity (which represents a completely reversible case).

Line(s) 314-317. Please discuss the meaning of the better fit to pseudo-second order adsorption kinetic model. Generally, it means the adsorption capacity is more dependent on the surface adsorbate amount.

Answer: It is the same meaning proposed by reviewer

Round 2

Reviewer 2 Report

I think this manuscript was significantly improved and can be published after minor revisions below.

Line(s) 201-203. Please put reference(s) about this statement(s).

Line(s) 336-338. Please put reference(s) about RL.

Line(s) 360-361. Please put reference(s) about this statement(s).

Figures. Pleas improve the resolutions, if possible.

Author Response

Line(s) 201-203. Please put reference(s) about this statement(s).

New reference 36 inserted

Line(s) 336-338. Please put reference(s) about RL.

New reference 48 inserted

Line(s) 360-361. Please put reference(s) about this statement(s).

Answer: New reference 50 added

Figures. Pleas improve the resolutions, if possible.

Answer: the figures resolution modified in the last version and it can be reduced or expanded into the text to adjust resolution.